# Damage-Associated Molecular Patterns, a Class of Potential Psoriasis Drug Targets

**DOI:** 10.3390/ijms25020771

**Published:** 2024-01-07

**Authors:** Yaqi Gao, Bishuang Gong, Zhenxing Chen, Jierong Song, Na Xu, Zhuangfeng Weng

**Affiliations:** Shenzhen Key Laboratory of Systems Medicine for Inflammatory Diseases, School of Medicine, Shenzhen Campus of Sun Yat-sen University, Shenzhen 518107, China; gaoyq36@mail.sysu.edu.cn (Y.G.); gongbishuang@hotmail.com (B.G.); chenzhx36@mail2.sysu.edu.cn (Z.C.); songjr7@mail2.sysu.edu.cn (J.S.)

**Keywords:** DAMPs, psoriasis, drug targets, autoimmune diseases, psoriatic inflammation, immune mechanism

## Abstract

Psoriasis is a chronic skin disorder that involves both innate and adaptive immune responses in its pathogenesis. Local tissue damage is a hallmark feature of psoriasis and other autoimmune diseases. In psoriasis, damage-associated molecular patterns (DAMPs) released by damaged local tissue act as danger signals and trigger inflammatory responses by recruiting and activating immune cells. They also stimulate the release of pro-inflammatory cytokines and chemokines, which exacerbate the inflammatory response and contribute to disease progression. Recent studies have highlighted the role of DAMPs as key regulators of immune responses involved in the initiation and maintenance of psoriatic inflammation. This review summarizes the current understanding of the immune mechanism of psoriasis, focusing on several important DAMPs and their mechanisms of action. We also discussed the potential of DAMPs as diagnostic and therapeutic targets for psoriasis, offering new insights into the development of more effective treatments for this challenging skin disease.

## 1. Introduction

Psoriasis is a chronic systemic inflammatory disease with a global incidence ranging from 0.09% to 11.4% [1]. From 1990 to 2019, the global prevalence of psoriasis rose by approximately 27%, with the number of affected individuals increasing from 3,653,236 to 4,622,594 [2]. Psoriasis can affect individuals of all ages, but young adults (around 20 years old) and elderly people (around 60 years old) are the two first peaks of onset [3,4]. The skin is the primary site of onset, and the most commonly affected areas include the elbows, knees, scalp, lumbar spine, and nails [5]. Pathological features such as epidermal thickening, dermal vasodilation, and inflammatory cell infiltration are often present at the lesion site [6]. Based on symptoms, psoriasis is mainly divided into three categories: psoriasis vulgaris, erythrodermic psoriasis, and pustular psoriasis. Psoriasis vulgaris, or chronic plaque psoriasis, accounts for 80–90% of patients [7]. However, the etiology and pathogenesis of psoriasis are not yet fully understood. The pathogenesis of psoriasis is complex, involving genetic and environmental factors.

It has been proven by Rajan P. Nair et al. that *HLA-Cw6* is a *PSORS1* risk allele that is highly associated with early onset psoriasis [8]. Importantly, unhealthy habits like smoking and alcoholism are key pathogenic contributors as well. With the development of immunology and genetics in recent decades, the abnormal proliferation of keratinocytes caused by immune regulatory disorders was considered to be the primary cause of psoriasis [9]. Furthermore, the IL-23/IL-17 axis was also found to be crucial in the immunological control of psoriasis [10]. The disfunction of the IL-23/IL-17 axis causes an imbalance in the secretion of inflammatory factors such as IL-23, IL-17A, and TNF, which leads to a cascade response of inflammatory factors in keratinocytes and ultimately progresses into psoriatic lesions. Conversely, blocking the inflammatory factors TNF, IL-23, IL-17A, or the corresponding receptor IL-17RA has been shown to be effective in the treatment of psoriasis [11,12,13,14].

In addition to the disorder of the IL-23/IL-17 axis, recent evidence suggests that endogenous molecules such as damage-associated molecular patterns (DAMPs) also play a crucial role in the immune regulation of psoriasis. DAMPs, known as danger signal molecules, are essential initiators of inflammatory responses and regulators of cytokine production and are mainly released by damaged or necrotic cells as well as activated immune cells. They can activate the cellular immune response by binding to pattern recognition receptors (PRRs) on immune cells, triggering a “sterile inflammation” response [15]. Several DAMP molecules, including high mobility group box 1 (HMGB1), uric acid, heat shock proteins (HSPs), ATP, the S100A family, and self-DNA, have been implicated in psoriasis pathology [16]. This review provides a comprehensive overview of the role and mechanism of DAMPs in the initiation and progression of psoriasis, exploring their potential as new targets for psoriasis diagnosis and treatment.

## 2. Immune Regulation in Psoriasis

Psoriasis is an autoimmune disease that involves both innate and adaptive immunity in its pathogenesis. The innate immune response in psoriasis is primarily mediated by dendritic cells, macrophages, and innate lymphocytes, while the adaptive immune response is mainly mediated by T cells, including helper T cells and regulatory T cells. The immune cells interact with the keratinocytes, causing the skin inflammation cycle in psoriasis.

### 2.1. Innate Immune Regulation in Psoriasis

Dendritic cells, including plasmacytoid DCs (pDCs), conventional DC1s (cDC1s), conventional DC2s (cDC2s), inflammatory DCs (iDCs), and Langerhans cells (LCs), are important antigen-presenting cells involved in innate immunity in psoriasis [17]. pDCs can recognize DAMPs, such as autologous nucleic acids released by keratinocyte injury, antimicrobial peptides, or viral nucleic acids, through TLR7/9 receptors on the cell membrane, leading to the up-regulation of IFN-α expression [18,19]. IFN-α released by pDCs can activate local cDCs to secrete more cytokines and cause the differentiation of iDCs, which aggravates the early inflammatory response in psoriasis. Both pDCs and cDCs are derived from bone marrow and are more concentrated in damaged skin than in the undamaged skin of patients [20]. The activation of cDCs leads to the proliferation and polarization of downstream T cells and the production of cytokines such as IFN-γ and IL-17A, which accelerate the progression of psoriasis [21]. iDCs, derived from monocytes, can secrete IL-1β, IL-6, TGF-β, and IL-23. Downstream of iDCs, Th17 cells are the main adaptive immune cells in psoriasis and directly regulate keratinocytes during the disease. Cytokines secreted by iDCs play a key role in promoting the polarization of naive CD4+ T cells to Th17s and maintaining Th17 activity [22]. LCs are a special kind of DC cell that exists in the epidermis. In a healthy state, they function as antigen-presenting cells (APCs). When the skin is stimulated, it quickly presents T cells in the local lymph nodes [23].

Macrophages, an important component of innate immunity, are mainly divided into two subtypes: M1 and M2 [24]. M1 belongs to pro-inflammatory macrophages and can be formed by the polarization of M0 induced by TLR ligands or IFN-γ. M1 macrophages play a pro-inflammatory role by secreting inflammatory cytokines such as TNF-α and IL-6, which accelerate the exacerbation of psoriasis [25]. M2 belongs to anti-inflammatory macrophages and can be formed by the polarization of M0 induced by IL-4 or IL-13. M2 macrophages mount an anti-inflammatory response and repair damaged tissue when cell damage occurs [26]. Additionally, Yuzhu Hou et al. found that a large amount of IL-17A, IL-22, and IFN-γ could be produced after IL-23-induced M0. This type of macrophage is different from M1 and M2. When adoptively transferred into psoriasis mice modeled by IMQ, these macrophages induced more severe skin inflammation. This finding suggests that macrophages may directly regulate keratinocytes through IL-17A and other inflammatory factors, indicating their potential role in psoriasis [27].

Innate lymphoid cells (ILCs) are located in the mucosa and viscera of humans and mice and are an important part of innate immunity. Based on gene expression and cytokine secretion, ILCs are divided into three subtypes: ILC1s, ILC2s, and ILC3s [28]. ILC3s are the major subtype involved in psoriasis, and the quantity of ILC3s is dramatically elevated in the lesioned skin of psoriasis patients [29]. In addition to Th17, ILC3s are a major source of IL-17. Moreover, ILC3s have also been demonstrated to elicit psoriasis-like symptoms in mice by secreting IL-17A, IL-17F, and IL-22 [30].

### 2.2. Adaptive Immune Regulation in Psoriasis

Psoriasis is characterized by an imbalance of T cells in the adaptive immune system. Among them, helper T cells (Th cells) and regulatory T cells (Treg cells) play important roles in regulating the immune response in psoriasis.

Th1 cells are the main source of pro-inflammatory cytokines such as TNF-α, IFN-γ, and IL-2 in the peripheral blood and lesion areas of psoriasis patients, while Th2 cells mainly secrete the cytokine IL-4, which significantly increases in patients with erythrodermic psoriasis [31,32]. These findings suggest that both Th1 and Th2 cells are involved in the regulation of immune disorders in psoriasis. In addition, Th17 cells are a new type of Th cells that actively participate in inflammatory responses and secrete the key pro-inflammatory cytokine IL-17A, which has been found to be up-regulated in psoriasis lesions and synovial fluid [33,34]. IL-17A can also act on keratinocytes and promote their proliferation and differentiation [35,36,37]. Additionally, inflammatory factors such as IL-6 and IL-23 can stimulate Th17 proliferation and induce Th17 to release pro-inflammatory factors such as IL-17A, IL-17F, IL-22, and GM-CSF [38,39]. These inflammatory factors elicit downstream keratinocyte responses.

Regulatory T cells (Tregs) are important for maintaining immune balance and inhibiting immune cells. Tregs are categorized into thymus-derived regulatory T cells (tTregs) and peripherally derived regulatory T cells (pTregs), based on their origin. They can also be induced in vitro in naive CD4^+^ T cells by IL-2 and TGF-β [40]. Tregs highly express the transcription factor Foxp3 and possess the ability to regulate inflammatory diseases [41,42]. In psoriasis, the regulatory function of Tregs is significantly impaired within the damaged skin [43]. The differentiation of Tregs was inhibited by IL-6 that was produced by upstream APCs and can promote the differentiation of Th17s. Disruption of the balance between Tregs and Th17 cells is a hallmark of several autoimmune diseases, including multiple sclerosis (MS) and inflammatory bowel disease (IBD), and it also plays a significant role in the pathophysiology of psoriasis [44].

Accumulating evidence shows that, in addition to the immune cells, DAMPs are also involved in the immune regulation of psoriasis. As shown in Figure 1, at the initial stage of psoriasis, antigen-presenting cells, including DCs, macrophages, and others, are activated and produce cytokines, which stimulate the production of more cytokines by other immune cells [18,22]. As the disease progresses, Th17 and other cells secrete cytokines that stimulate keratinocytes, causing excessive proliferation of keratinocytes and thickening of the epidermis [37]. Damaged and necrotic keratinocytes release DAMPs such as their own DNA antimicrobial peptides S100A8 and S100A9, which further activate pDCs and form a pro-inflammatory feedback loop [45,46]. Throughout the process, immune cells and keratinocytes interact with one another via various cytokines to produce a complex inflammatory pathogenesis, in which DAMPs play a significant role in immune control. Therefore, studying DAMPs is crucial for understanding the mechanisms behind psoriasis.

## 3. DAMPs and Inflammation

When tissues are damaged or cells die, immune-stimulatory products called DAMPs are released. These DAMP molecules can amplify inflammation by recruiting and activating immune cells and promoting the secretion of inflammatory factors. The ways and types of DAMPs released can vary depending on the type of cell damage and death. Necrosis, apoptosis, pyroptosis, or ferroptosis all can affect the types of DAMPs released [47]. DAMPs can be exposed on the plasma membrane, such as calreticulin, or secreted into the extracellular environment, such as HMGB1 and IL-1α. Other DAMPs, such as ATP and DNA, are produced at the end of cell death. Although DAMPs and pathogen-associated molecular patterns (PAMPs) have similar effects, their sources are essentially different. PAMPs are microbial products, such as lipopolysaccharides, while DAMPs are derived from endogenous molecules in the body, which mediate “sterile inflammation” [48]. Additionally, PAMPs can induce the production of DAMPs during infection, which in turn leads to more severe inflammation. For example, Derek S Wheeler et al. demonstrated that respiratory syncytial virus (RSV) infection results in the release of extracellular Hsp72 from the airway epithelium, which acts as a danger signal or alarm factor, causing the recruitment and activation of neutrophils in the lung [49].

As shown in Figure 2, DAMPs can bind to pattern recognition receptors (PRRs) on immune cells, including Toll-like receptors (TLRs) and nucleotide-binding oligomerization domain-like receptors (NLRs). In addition, some atypical PRRs, such as the receptor for advanced glycation end products (RAGE), can also recognize DAMPs, leading to the activation of subsequent pathways and promoting host inflammatory responses [50].

### 3.1. DAMP Regulates Inflammatory Response through TLRs

TLRs are a group of crucial receptors belonging to PRRs. Among them, TLR3, TLR7, and TLR9 mainly recognize self-nucleic acid, while TLR2 and TLR4 primarily respond to released extracellular matrix components [51]. TLRs are widely distributed and expressed not only on immune cells such as macrophages, dendritic cells (DCs), B cells, and neutrophils, but also on non-immune cells such as fibroblasts, epithelial cells, and keratinocytes [52]. Upon activation, TLRs recruit TIR-domain-containing adaptors such as myeloid differentiation primary response protein 88 (MyD88), TIR domain-containing adaptor protein/MyD88 adapter-like (TIRAP/MAL), TIR domain-containing adaptor-inducing interferon-beta (IFN-β)/TIR domain-containing adaptor molecule 1 (TRIF/TICAM1), and TIR domain-containing adaptor molecule/TRIF-related adaptor molecule 2 (TRAM/TICAM2) [53]. All TLR signaling pathways ultimately activate the transcription factor nuclear factor-κB (NF-κB), which controls the expression of various inflammatory cytokine genes [54]. Fang Ren et al. demonstrated that psoriasis-like inflammation can impair renal function through the TLR/NF-κB signaling pathway. Thus, inhibiting the expression of TLR/NF-κB-related proteins may be a promising strategy for treating renal injury caused by psoriasis [55].

### 3.2. DAMP Regulates Inflammatory Response through NLRs

The NOD-like receptor (NLR) family comprises several members, including NLRP1, NLRP3, NLRC4, and NAIP, among which NLRP3 is involved in the assembly of the inflammasome and is primarily responsible for sensing and responding to DAMPs [56]. DAMPs trigger abnormal ion flux, mitochondrial damage, and lysosomal rupture in cells, leading to the formation of the inflammasome [57]. This protein complex is composed of NLRP3, ASC, and pro-caspase1 [58]. Once assembled, the inflammasome releases activated caspase1, which cleaves pro-IL-1β and pro-IL-18, thereby releasing the inflammatory cytokines IL-1β and IL-18 and promoting immune-related inflammation [56]. NLRs are mainly expressed in immune cells, such as neutrophils, monocytes, and macrophages [52]. Studies have demonstrated that miR-155 silencing can inhibit the inflammatory response in psoriasis by regulating NLRP3 inflammasome activity [59].

### 3.3. DAMP Regulates Inflammatory Response through RAGE

The receptor for advanced glycation end products (RAGE) is a member of the immunoglobulin (Ig) superfamily of cell surface receptors that is expressed on nearly all cells and is upregulated in inflammation-related diseases [60]. Upon binding of DAMPs to RAGE, downstream signaling pathways are activated, including Ras and Cdc42/Rac, which subsequently activate stress-activated protein kinase/c-Jun-NH2 terminal kinase (SAPK/JNK) and mitogen activated protein kinase (MAPK) pathways. This activation ultimately leads to the activation of transcription factors NF-κB and the expression of downstream inflammatory factors [61]. In psoriasis, for instance, S100A7 was found to induce the expression of mature interleukin-1α via the RAGE-p38 MAPK-calpain 1 pathway, which triggers an inflammatory response in the body [62].

In addition to RAGE, other receptors such as C-type lectin receptors (CLRs), RIG-I-like receptors (RLRs), and TREMs also recognize DAMPs [63,64,65]. Most of these receptors are distributed on immune cells and activate related inflammatory pathways after binding with DAMPs, inducing inflammatory responses.

## 4. DAMPs in Psoriasis

DAMPs have a dual function. On the one hand, they can regulate cell homeostasis and maintain cell function when present within cells. On the other hand, they can also act as endogenous molecules of cell death or injury and amplify the signal of inflammation through various cell receptors upon release. This leads to the activation of immune cells and the secretion of a large number of inflammatory factors. DAMPs not only play a pro-inflammatory role in acute inflammation, such as sepsis, acute liver injury, or acute pancreatitis, but they also mediate immunity in chronic immune diseases such as rheumatoid arthritis, multiple sclerosis, and inflammatory bowel disease [66,67,68,69,70,71]. In recent years, extensive immunological research has confirmed the pro-inflammatory effects of DAMPs in various inflammation-related diseases. Consequently, these molecules are expected to serve as useful biomarkers and therapeutic targets for treating such diseases.

### 4.1. HMGB1 and Psoriasis

HMGB1 is a highly conserved non-histone nucleoprotein with a size of approximately 30 kDa [72]. It comprises two cognate DNA-binding domains and a negatively charged C-terminal domain [73]. HMGB1 is present in all higher eukaryotic cells and assists in DNA replication, transcription, and repair by binding to chromatin in the nucleus [74]. Importantly, it also functions as a DAMP in infection, inflammation, and immune processes, primarily by stimulating TLR2, TLR4, and RAGE receptors. In recent years, increasing evidence suggests that HMGB1 is closely linked to psoriasis. For instance, researchers have detected a significant elevation of serum HMGB1 levels in psoriasis patients compared to healthy individuals, with levels being closely related to disease severity [75]. In addition, the expression of HMGB1 in the lesion sites of psoriasis patients was also significantly increased compared with healthy people [76].

HMGB1 is involved in psoriasis progression through several mechanisms. A study by Zhen Wang et al. revealed that HMGB1 promotes the expression of psoriasis-associated cytokines and antimicrobial peptides by activating autophagy in keratinocytes. HMGB1 also amplifies the IL-23/IL-17 immune cycle by activating dermal γδ T cells to produce IL-17A [77]. Furthermore, HMGB1 upregulates the expression of TLR2, TLR4, and RAGE in the skin lesion area of psoriasis and activates the inflammasome and NF-κB pathway in keratinocytes to promote IL-18 secretion, which would aggravate the condition of IMQ model psoriasis mice [78]. In addition, Lisa Strohbuecker et al. conducted a comprehensive study on the relationship between HMGB1 and immune cells, revealing that HMGB1 expression was markedly elevated in the psoriatic skin lesions, and its receptor RAGE expression was also significantly increased on CD8^+^ cells and CD4^+^ Treg cells within the skin lesion area, which was strongly associated with the disease progression in psoriasis [79]. Collectively, HMGB1 plays a crucial role in both downstream keratinocytes and upstream immune cells.

The role of HMGB1 in psoriasis has been extensively studied, and its potential as a therapeutic target has been demonstrated in various preclinical studies. HMGB1 knockdown or inhibition using antibodies or small molecules have shown significant improvements in psoriasis symptoms in animal models. For example, *Hmgb1*-KD mice and mice treated with HMGB1 antibodies exhibited improved disease outcomes after IMQ-induced psoriasis modeling [77]. Additionally, treatment with the HMGB1 inhibitor glycyrrhizin alleviated the condition of IMQ psoriasis mice [78]. The mechanism of action of HMGB1 blockade involves the inhibition of Th17 cell responses, the reduction in γδ T cells, and the regulation of inflammatory cytokine expression, such as IL-6, TNF-α, IFN-γ, and IL-17, leading to the suppression of clinical and histological evolution of IMQ-treated skin [78,80]. Interestingly, the mechanism of action of HMGB1 has also been implicated in existing psoriasis drugs, such as methotrexate (MTX), which inhibits the interaction between HMGB1/RAGE by binding to the RAGE binding region of HMGB1 [81]. Overall, HMGB1 is an important DAMP in the pathogenesis of psoriasis and a promising diagnostic and therapeutic target.

### 4.2. S100 Protein and Psoriasis

The S100 protein family consists of 25 members that share a molecular weight of about 10–12 kDa and maintain an amino acid sequence similarity of 25–65% [82,83]. Most S100 proteins are Ca^2+^ signaling proteins that contain a conserved calcium-binding motif called EF-hand. Upon binding to Ca^2+^, S100 proteins interact with other proteins to regulate a range of cellular functions, including cell migration, proliferation, apoptosis, and maintenance of Ca^2+^ homeostasis [84]. When released outside the cell as DAMPs, S100 proteins play a pro-inflammatory role by binding to receptors such as TLRs and RAGE [82]. In psoriasis, S100A7, S100A8, S100A9, S100A12, and S100A5, defined as “antimicrobial peptides”, are the main S100 family members involved. They are released by keratinocytes and activate innate immunity [85]. Among them, S100A7 is overexpressed in the damaged skin of psoriasis patients but tends to decrease in serum with increasing disease severity [86]. In contrast, both S100A8 and S100A9 are significantly increased in the serum and damaged skin of psoriasis patients [87].

The role of S100 family proteins in regulating psoriasis is still controversial. Hu Lei et al. demonstrated that human S100A7 aggravates psoriasis severity by inducing the expression of interleukin 1α in mature epidermal keratinocytes through the RAGE-p38 MAPK-calpain 1 pathway [62]. In a model of psoriasis induced by IMQ, Joan Defrêne et al. found that S100A8 inhibits the proliferation of keratinocytes and regulates their differentiation [88]. However, some scholars suggest that S100 family proteins are mainly secreted by keratinocytes and act on immune cells. Carolin Christmann et al. found that the expression of S100A8/S100A9 did not significantly affect the maturation and inflammatory response patterns of keratinocytes in primary *S100a9^−/−^* keratinocytes, indicating that keratinocytes are not the target cells for the pro-inflammatory effect of S100A8/S100A9 [89].

The potential of S100s as targets for psoriasis diagnosis and treatment has been suggested. Henry J. Grantham et al. proposed that S100A8/9 in serum can serve as a biological indicator of atherosclerosis caused by psoriasis [90]. Helia B. Schonthaler et al. identified S100A9 as a chromatin component that regulates C3 expression in mouse and human cells by binding to a region upstream of the C3 initiation site. Knocking out S100A9 in a psoriasis mouse model strongly attenuated psoriasiform skin disease and inflammation, with reduced C3 volume and mild immune infiltration [91]. However, the study by Joan Defrêne et al. showed that S100A8 and S100A9 knockout mice had more severe symptoms after IMQ-induced psoriasis modeling, with increased IL-17 responses in the dermis and lymph nodes, indicating the anti-inflammatory properties of S100A8 and S100A9 [88]. Overall, while the S100 family is involved in regulating multiple immune pathways in psoriasis, its main role requires further investigation.

### 4.3. HSP and Psoriasis

Heat shock proteins (HSPs) are a ubiquitous class of proteins that are synthesized in response to conditions such as heat shock, heavy metals, infection, or an abnormal physiological state. They have intracellular and extracellular functions. Intracellular HSPs are involved in protein folding, recognizing and binding nascent polypeptide chains and partially folded intermediates of proteins to prevent their aggregation and misfolding [92]. In contrast, extracellular HSPs function as DAMPs. Among them, HSP60, HSP70, and HSP90 are mainly involved in psoriasis and act through receptors such as TLR2, TLR4, and CD91 [93,94]. Studies have shown that the average IRID (immunoreactivity intensity distribution index) score of HSP60 expression in the basal, suprabasal, and superficial epidermal layers of psoriatic skin is higher than that of healthy skin [95]. HSP70 is also expressed in the skin of psoriasis sufferers but primarily in the basal layer [96]. HSP90 expression increases throughout the year with the frequency of psoriasis exacerbations, and psoriasis-related hyperlipidemia or diabetes also upregulates HSP90 expression [97]. These studies have shown that the expression levels of HSP family proteins are closely related to the severity of psoriasis.

Recent studies have shed light on the mechanisms underlying the involvement of HSP family proteins in the regulation of psoriasis. For example, O. Boyman et al. reported that dendritic antigen-presenting cells expressing the HSP receptor CD91 accumulate during the occurrence of psoriasis lesions, accompanied by activation of NF-κB and increased production of TNF. This suggests that HSP family proteins may affect the migration of dendritic cells through CD91 [98]. Furthermore, Jonathan L. et al. demonstrated that HSP can induce the maturation of blood-derived DCs, accompanied by increased IL-12 production and enhanced antigen presentation function [99]. The anti-inflammatory pain drug luteolin can regulate the ratio of immune cells by inhibiting the expression of HSP90 and exosome secretion and relieving the lesions and symptoms of psoriasis [100]. In addition, some drugs treat psoriasis by targeting the MAPK pathway and inhibiting the expression of HSP90 and HSP60 in keratinocytes [101]. HSP90 inhibitors have been shown to significantly improve psoriasis symptoms in a xenograft mouse model [102]. Specifically, Rikke S. Hansen et al. demonstrated that HSP90 inhibitors can reduce the expression of pro-inflammatory cytokines IL-23, IL-6, and TNF caused by activation of TLR3 in keratinocytes, leading to a reduction in the inflammatory response [103].

### 4.4. Other DAMPs and Psoriasis

Other DAMPs, such as cytosolic DNA, ATP, self-RNA, the LL37 complex, and IL-33, are also implicated in psoriasis. For instance, the release of cytosolic DNA as DAMP induces the synergistic activation of STING in immune cells and keratinocytes, amplifying the inflammatory response in psoriasis [104]. ATP released from keratinocytes triggers the Koebner phenomenon in psoriasis [105]. The self-RNA and LL37 complex can activate DC to generate inflammatory cytokines and aggravate psoriasis [106]. IL-33 can worsen the condition of IMQ psoriasis model mice, reduce CD4 + T and CD8 + T cells in the spleen of psoriatic mice, inhibit autophagy in the skin, and promote STAT3 tyrosine phosphorylation [107]. In summary, despite the critical roles of various DAMPs in regulating inflammation in psoriasis (Table 1), further studies are required to explore their specific functions and mechanisms.

## 5. DAMPs Regulate the Immune Mechanism of Psoriasis

Psoriasis is an autoimmune disorder characterized by abnormal immune responses involving both innate and adaptive immunity. The immune system secretes inflammatory factors that act on keratinocytes, while damaged keratinocytes release DAMPs that in turn react on immune cells, triggering an inflammatory response circuit, which makes the course of psoriasis difficult to heal. DAMPs play a crucial role in the immune pathogenesis of psoriasis, including the following: (1) activation of innate immunity—regulating the secretion of upstream inflammatory factors such as IL-1 and IL-23 by acting on antigen-presenting cells (APCs); (2) regulation of adaptive immunity—amplifying inflammatory signals by acting on T cells, particularly Th17 cells; and (3) induction of physiological dysfunction of keratinocytes—forming the disease phenotype by acting on keratinocytes. In this microenvironment, keratinocytes release more DAMPs, perpetuating the cycle of inflammation.

### 5.1. Activation of Innate Immunity

DCs, macrophages, and neutrophils are the main innate immune cells. These cells express PRRs and also release DAMPs. Previous studies have demonstrated that DAMPs are involved in various inflammatory diseases, including psoriasis, multiple sclerosis, inflammatory bowel disease, systemic lupus erythematosus, and arthritis, by acting on APCs. In psoriasis, DAMPs can directly impact the function of APCs. For example, HMGB1 can promote DC migration and maturation through RAGE, and S100 family proteins can promote the migration, expansion, and differentiation of macrophages [109,110,111]. Some S100 family proteins can also act as neutrophil extracellular trap (NET) release markers that activate neutrophils [112]. Moreover, DAMPs can activate APCs to promote the secretion of inflammatory factors such as IL-6 and IL-23. They also promote an increase in the proportion of Th17 cells or the secretion of inflammatory factors such as IL-17A and IL-17F. For instance, ATP can act as a DAMP to drive DCs to promote Th17 differentiation, which can exacerbate the psoriasis process [113].

### 5.2. Regulation of Adaptive Immunity

DAMPs have been shown to directly regulate T cells, which are an important adaptive immune cell type in psoriasis. These immune cells express various receptors that recognize DAMPs. For example, knocking out the HSP90 receptor ACT on T cells results in disturbed T cell function, leading to an excessive Th17 response. Th17 cells are important in psoriasis because they express various TLRs on their membrane [114]. Activation of TLR2 receptors can stimulate the differentiation of naïve CD4^+^ T cells into Th17 cells, and stimulating TLR2 and TLR4 on mature Th17 cells will increase the secretion of IL-17A [115]. Studies have also demonstrated that activating TLR4 receptors can promote the differentiation of naïve CD4^+^ T cells into Th17 cells in the EAE model [116]. Furthermore, DAMPs are known to have a regulatory effect on Treg, with HSP60 and HSP70 speculated to induce Treg differentiation [117]. Taken together, these findings demonstrate that DAMPs have a multifaceted regulatory effect on immunity.

### 5.3. Induction of Physiological Dysfunction of Keratinocytes

DAMPs have a direct effect on the psoriatic phenotype via regulating keratinocytes. When acting directly on keratinocytes, DAMPs can result in dysregulation of keratinocyte differentiation and can influence the production of keratinocyte cytokines. For instance, through the RAGE-p38 MAPK-calpain 1 pathway, S100A7 can stimulate the expression of IL-1α in psoriatic keratinocytes [62]. Moreover, keratinocytes can produce more DAMPs when stimulated by inflammatory factors. During psoriasis pathogenesis, the epidermal microenvironment is enriched with inflammatory factors such as IL-17A and IL-17F, which can induce the expression of DAMPs in keratinocytes [89].

Taken together, DAMPs regulate the immune cycle of psoriasis by activating innate immunity and regulating adaptive immunity, as well as causing physiological dysfunction of keratinocytes. From the upstream immune cells to the downstream keratinocytes, the effects of DAMPs on target cells and the inflammatory signals they mediate vary due to the diverse types and sources of DAMPs. This diversity in the DAMP-regulated inflammatory response network may be a key factor contributing to different clinical manifestations of psoriasis.

## 6. Future Directions and Prospects for Diagnosis and Treatment

In recent years, significant progress has been made in the diagnosis and treatment of psoriasis, largely due to the advent of biological agents. For instance, antibodies targeting IL-23 and IL-17, such as Guselkumab and Secukinumab, have greatly promoted the clinical management of psoriasis [12,118]. DAMPs play a crucial role in the body’s immune response, and numerous drugs have been developed to target them. Currently, the development of diagnostic and therapeutic agents for DAMPs is primarily focused on three key directions: (1) utilizing DAMP as a diagnostic marker, (2) directly inhibiting or using antibodies against DAMP, and (3) inhibiting DAMP receptors or signaling pathways.

### 6.1. Utilizing DAMP as a Diagnostic Marker

During the course of psoriasis, DAMPs are significantly up-regulated and decrease with the relief of symptoms after treatment, making them a potential diagnostic marker. Various DAMPs, including S10015, have been found to increase during the onset of psoriasis. The expression of S10015 in PBMCs of psoriasis patients is higher than that of healthy individuals, and its level is inhibited after effective treatment with narrow-band UVB [119]. This suggests that S10015 is highly correlated with the severity of psoriasis and has the potential to serve as an auxiliary diagnostic marker in clinical practice. However, the disease specificity of most DAMPs for clinical diagnosis has not been fully confirmed, and further exploration is needed to develop diagnostic markers for DAMPs.

### 6.2. Directly Inhibiting or Using Antibodies against DAMP

Recent research studies showed that direct inhibition or antibodies against DAMP have emerged as a potential therapeutic strategy for psoriasis. In vivo experiments have demonstrated that knockout of DAMPs or treatment with neutralizing antibodies can significantly alleviate psoriasis symptoms. Inhibitors that directly target DAMPs, such as glycyrrhizin and diflunisal (DFL), which inhibit HMGB1, can also significantly improve the condition of IMQ-modeled psoriasis mice [120]. Furthermore, compound glycyrrhizin has been shown to synergize with traditional treatments, effectively increasing the number of patients achieving PASI60 and PASI90 [121]. Currently, numerous drugs that target DAMPs are undergoing clinical trials for the treatment of psoriasis. One such drug is RGRN-305, an HSP90 inhibitor initially tested as an anti-tumor drug that unexpectedly improved psoriasis symptoms. RGRN-305 is currently undergoing Phase 1 evaluation in patients with moderate to severe psoriasis [103]. Although direct inhibition or antibodies against DAMP hold great promise as a therapeutic approach, further research is needed to fully evaluate its effectiveness and potential side effects.

### 6.3. Inhibiting DAMP Receptors or Signaling Pathways

Blocking DAMP receptors or signaling pathways is an important strategy for developing diagnostic and therapeutic reagents targeting DAMPs. For instance, Soluble RAGE (sRAGE) can bind to HMGB1, thereby breaking the interaction of HMGB1 with RAGE and weakening its stimulating effect. This can limit the proliferation and expansion of liver cancer cells [122]. Similarly, Triptolide has been shown to inhibit the HMGB1/TLR4/NF-κB pathway, thereby inhibiting breast cancer [123]. Additionally, Calcimycin has been shown to target the expression of S100A4 to limit the motility of colon cancer cells [124]. These studies provide promising ideas for the development of DAMP-targeting treatments for psoriasis.

Currently, antibodies targeting IL-17A and IL-23 are widely used for the treatment of psoriasis. However, some patients may not respond to these treatments due to physical reasons, leading to treatment failure or secondary treatment failure due to the attenuation of efficacy over time. Furthermore, many antibody-based biologics are quite expensive, making it challenging for ordinary patients to maintain long-term treatment. As DAMPs are an emerging Immune target, the development of drugs targeting DAMPs may offer more treatment options for patients and potentially become a breakthrough for psoriasis treatment.

## 7. Conclusions

Throughout the development of therapeutic drugs for psoriasis, biological targeting agents have become the main treatment, with a better understanding of the mechanisms of psoriasis. There has been a shift from addressing skin discomfort to addressing the underlying problem of psoriasis—immune disorders. As a chronic immune disease, psoriasis involves a complex immune regulation mechanism. Here, we have provided a detailed introduction to the immunoregulatory network of psoriasis and summarized the roles and mechanisms of various DAMPs in psoriasis. Psoriasis is centered around the IL-23/IL-17 axis. Upon activation of innate immunity, it secretes inflammatory factors that affect Th17 cells, γδT cells, or Treg cells. At the same time, under the stimulation of abundant inflammatory factors, downstream keratinocytes further produce more DAMPs, which act on immune cells. In this process, DAMPs act as danger signals and run through the entire psoriasis immune cycle. Moreover, many studies have shown that DAMPs can be used as targets for the diagnosis and treatment of psoriasis, achieving promising results in clinical tests and in vivo and in vitro experiments. However, the precise functions and immune mechanisms of some DAMPs in psoriasis are not fully understood. More clinical research data are needed to support the development of DAMP as a diagnostic and therapeutic agent for psoriasis. In conclusion, the role of DAMPs in psoriasis should not be underestimated, and further research is necessary to fully explore their potential as therapeutic targets.

## Figures and Tables

**Figure 1 ijms-25-00771-f001:**
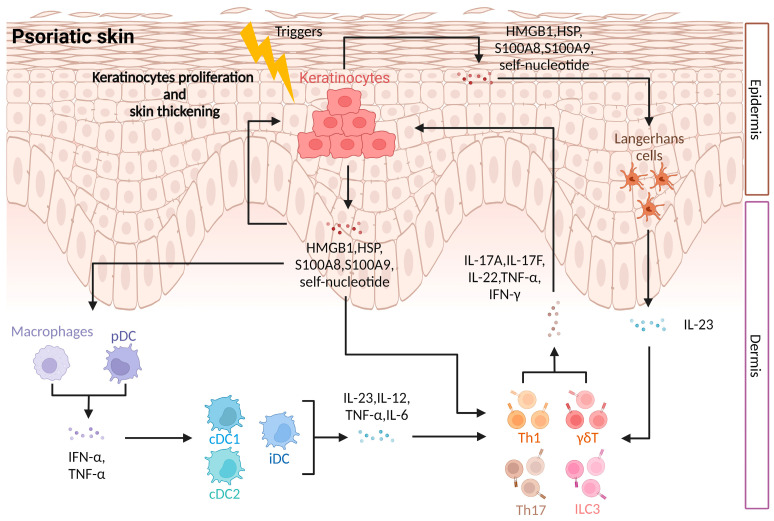
Immune regulation in psoriasis. Activated APCs produce cytokines such as TNF-α, IL-6, IL-12, and IL-23. Th17 and other cells secrete IL-17A, IL-17F, and IL-22 under the stimulation of inflammatory factors. These cytokines stimulate keratinocytes, leading to hyperproliferation of keratinocytes and thickening of the epidermis. Keratinocyte injury and necrosis release DAMPs, and these endogenous molecules further activate APCs. (Created with Biorender.com (accessed on 5 December 2023)).

**Figure 2 ijms-25-00771-f002:**
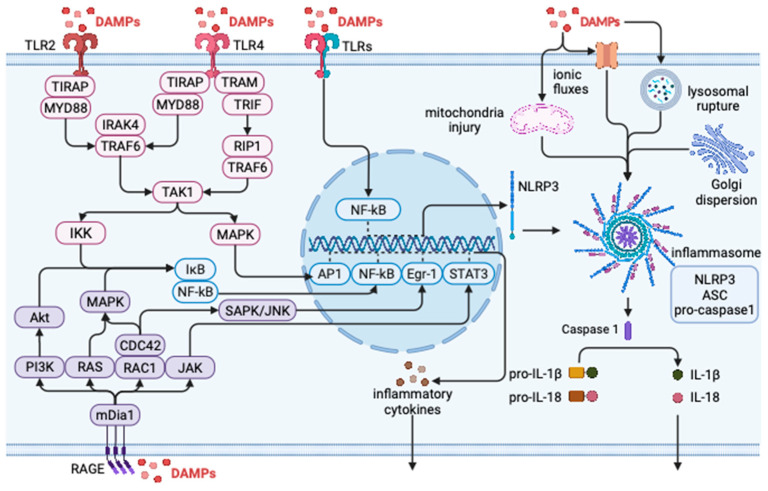
DAMPs in the inflammatory response. DAMPs stimulate TLR receptors such as TLR2 and TLR4; recruit MyD88, TIRAP, TRAM, TRIF, and other adapters; and finally activate transcription factors such as AP-1 and NF-kB to produce inflammatory factors. DAMPs stimulate RAGE receptors, activate MAPK and SAPK/JNK pathways, and finally activate transcription factors such as STAT3 and NF-kB to produce inflammatory factors. DAMP-induced ion fluxes, mitochondrial injury, and lysosomal rupture lead to the assembly of inflammasomes. The assembled inflammasome releases caspase 1. Caspase 1 will cleave pro-1β and pro-18 to produce mature IL-1β and IL-18. (Created with Biorender.com (accessed on 5 December 2023)).

**Table 1 ijms-25-00771-t001:** The role of DAMP in psoriasis.

DAMP	Psoriasis Patients	Experimental Psoriasis
HMGB1	It is increased in the serum and skin of psoriasis patients [76,79].	Its inhibitor, antibody, or deficiency reduces inflammation of IMQ-induced mice [77,78,80].
S100s	S100A7: It is strongly expressed in psoriatic lesions and decreased in the serum of patients with psoriasis [86]. S100A8/A9: It is increased in the serum and lesion skin of psoriasis patients [87].	S100A8/A9: Serum S100A8/A9 levels may act as biomarker of atherosclerosis severity in psoriasis; deletion of them enhances inflammation in imiquimod-induced psoriasis mice [88,90].
HSPs	HSP60: The mean IRIDI scores for its expression in the basal, suprabasal, and superficial epidermal layers of psoriasis are higher than those of normal skin [95]. HSP70: The expression of it has no obvious differences between lesion and normal skin [96]. HSP90: The expression of it increases with the frequency of exacerbations of psoriasis throughout the year [97].	HSP70: It prevents imiquimod-induced, psoriasis-like inflammation in mice [108]. HSP90: Its inhibitor alleviates psoriasis in a xenograft transplantation model [103].

## Data Availability

There is no new data were created.

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
