# Peer review of "Damage-Associated Molecular Patterns, a Class of Potential Psoriasis Drug Targets"

_ijms, 2024, doi:10.3390/ijms25020771_

Round 1

Reviewer 1 Report

Comments and Suggestions for Authors

The authors have managed to comply a comprehensive review about DAMPs involvement in psoriasis patogenesis and treatment. The detailed description of immune processes underlying various aspects of DAMP influence of adaptive and innate immune responses related psoriasis inflammation maintenance are provided. The mainstream of the review seems very promising in future investigations of psoriasis pathogenesis understanding and antipsoriasis drug development. Several flaws however should be corrected prior to consideration of this review for publication. First, the authors mentioned about the psoriasis incidence but no figures provided about the morbidity changing. Please, indicate if psoriasis incidence rising of decreasing recent times. The term 'racial differences' (line 26) is still better to avoid, considering that this psoriasis peculiarity is not discussed in manuscript content. The diagnosis 'arthritic psoriasis' (lines 33-34) can be completely different from psoriasis: the correct term, encompassing both conditions, is 'psoriatic disease'. Besides, that is the reason the authors mentioned 'four categories'? Does it assume peculiar DAMP involvement depending on psoriasis symptomes?

Reviewer 2 Report

Comments and Suggestions for Authors

This is an extensive and well-prepared review. I have several minor comments, which are described below.

Lines 27- 28 Please clarify: “young adults (around 20 years old) and middle-aged and elderly people (around 60 years old) are the two first peaks of onset middle-aged or elderly people”. Which are the 2 peaks?

Line 33: I would mention “psoriasis vulgaris or chronic plaque psoriasis”

Line 38: I would not include infections as an unhealthy habit.

Line 162: explain the abbreviation PAMP

Line 165: Please explain the abbreviation RSV

Line 295: please reformulate: “psoriasis sickness” (maybe the psoriasis severity)

Table 1: I prefer the term serum instead of sera

Comments on the Quality of English Language

I have doubts about the word "sera". I think it is more appropriate to use the word "serum".
